# Role of comprehensive geriatric assessment in healthcare of older people in UK care homes: realist review

Neil H Chadborn,[1,2] Claire Goodman,[3,4] Maria Zubair,[5] Lídia Sousa,[6] John R F Gladman,[1,2,7] Tom Dening,[8] Adam, L Gordon[2,7,9,10]

For numbered affiliations see end of article.

**Correspondence to**
Dr Neil H Chadborn;
Neil.Chadborn@nottingham.ac.uk

## ABSTRACT

**Objectives** Comprehensive geriatric assessment (CGA) may be a way to deliver optimal care for care home residents. We used realist review to develop a theory-driven account of how CGA works in care homes.

**Design** Realist review.

**Setting** Care homes.

**Methods** The review had three stages: first, interviews with expert stakeholders and scoping of the literature to develop programme theories for CGA; second, iterative searches with structured retrieval and extraction of the literature; third, synthesis to refine the programme theory of how CGA works in care homes. We used the following databases: Medline, CINAHL, Scopus, PsychInfo, PubMed, Google Scholar, Greylit, Cochrane Library and Joanna Briggs Institute.

**Results** 130 articles informed a programme theory which suggested CGA had three main components: structured comprehensive assessment, developing a care plan and working towards patient-centred goals. Each of these required engagement of a multidisciplinary team (MDT). Most evidence was available around assessment, with tension between structured assessment led by a single professional and less structured assessment involving multiple members of an MDT. Care planning needed to accommodate visiting clinicians and there was evidence that a core MDT often used care planning as a mechanism to seek external specialist support. Goal-setting processes were not always sufficiently patient-centred and did not always accommodate the views of care home staff. Studies reported improved outcomes from CGA affecting resident satisfaction, prescribing, healthcare resource use and objective measures of quality of care.

**Conclusion** The programme theory described here provides a framework for understanding how CGA could be effective in care homes. It will be of use to teams developing, implementing or auditing CGA in care homes. All three components are required to make CGA work—this may explain why attempts to implement CGA by interventions focused solely on assessment or care planning have failed in some long-term care settings.

**Trial registration number** CRD42017062601.

## INTRODUCTION

Around 425 000 people live in care homes in the UK.[1] Care homes are classified as either care homes with, or without, nursing based

### Strengths and limitations of this study

► Realist methods enabled us to explore the contextual factors and underpinning causal mechanisms associated with using comprehensive geriatric assessment (CGA) in care homes.

► The review makes transparent how CGA is thought to work and demonstrates what is supported by the evidence.

► We were limited to the detail included in the studies retrieved, which provided more detail about assessment than about care planning or management.

on the availability of registered nurses on-site. The types of residents cared for in both classifications of facility are similar and all UK care homes are included in the international consensus definition of a nursing home.[2] Care home residents are often frail and characteristically have complex medical histories, multiple morbidities and high levels of functional dependency.[3] UK care homes are primarily institutions that meet the basic and personal care needs of people with stable disabilities, but rely to varying extents on external healthcare support to meet residents' needs such as symptom control in long-term conditions, end-of-life care and the consequences of frailty such as delirium and falls. Challenges in meeting their healthcare needs include variable access to healthcare professionals, inadequate coordination of multiple teams providing care, failure to harness the expertise of care home staff and inappropriate access to expertise in dementia care.[4 5]

One possible response to these deficiencies has been proposed to be comprehensive geriatric assessment (CGA).[6] Rubenstein *et al* defined CGA as a 'multidisciplinary diagnostic process intended to determine a frail older person's medical, psychosocial and functional capabilities and limitations in order to develop an overall plan for treatment and

long-term follow-up'.[7] CGA has been shown to improve outcomes for older people in hospital[8–10] and community settings.[11] It encompasses health and social care needs and facilitates multidisciplinary working. However, evidence regarding its effectiveness in care homes is limited,[12] and there are limited data describing what needs to be in place for uptake and sustained implementation in this setting.

Although CGA has been identified to have three main components—structured comprehensive assessment, developing a plan of care and working towards the delivery of patient-centred goals[13]—how these components work to achieve improved patient outcomes has not been described. It is also not clear if CGA works in the same way in care homes as it does in hospital settings, from where most of the literature derives.

Descriptions of CGA range from solely an assessment process,[14] to a model of interprofessional working and integrated care across health and social care.[15] In care homes, national and local policy shapes how CGA is used. Some countries mandate use of a systematised assessment and care planning algorithm in the care home sector, with the US Minimum Dataset (MDS)[16] and the widely used International Resident Assessment Instrument (InterRAI)[17] being the most frequently cited examples. There is evidence that these can provide a basis for CGA.[18] Very little has been written about what contextual factors support delivery of CGA in care homes and how CGA operates to improve standards of care.

### Objectives of realist review

This realist review explored the contexts, mechanisms and outcomes which enable CGA to provide a systematic approach to assessment and care delivery. Realist synthesis methodology was chosen because it is designed to identify and understand the causal processes and contexts which explain the outcomes of complex interventions.[19] We aimed to provide a theory-driven explanation of how CGA works in care homes.

To provide a focus for our review, we focused initially on health-related quality of life (HRQoL), and satisfaction with, and appropriate use of, health and social care services. These were chosen because they represented the priorities of service users and providers.[20 21]

The specific objectives of the realist review were:
1. To identify and characterise the elements which make CGA useful as a way of organising healthcare in care homes, under what circumstances and why.
2. To understand the configuration of different contexts in which CGA is implemented and how these may act as a resource, or trigger particular mechanisms, to achieve the successful implementation, uptake and working of CGA in care homes.
3. To seek evidence on the feasibility of using CGA within UK care home settings and its resource implications and costs.

## METHODS
The full protocol for the Proactive Healthcare in Care Homes (PEACH) realist review has been published elsewhere.[22] We have adhered to the Realist And Meta-narrative Evidence Syntheses: Evolving Standards guidelines for reporting realist reviews[23] (see online supplementary file 1 for checklist). Databases were searched initially on 19 May 2017 and finally on 12 July 2018.

A three-stage approach was used.[23] Stage 1 was designed to develop an explanatory model which would provide an organising framework for the rest of the review. This comprised an initial scoping review of the literature and also stakeholder interviews to identify possible mechanisms and outcomes. Search terms and inclusion and exclusion criteria for the review are summarised in the full published protocol and were chosen to identify literature which used the term 'Comprehensive Geriatric Assessment' (CGA) to describe comprehensive assessment and management of care home residents. The only deviation from the published protocol was that we did not use the Medical Subject Heading (MeSH) term 'Geriatric Assessment' in PubMed. This was because we found this term to be highly inclusive, returning articles on topics ranging from posture and mobility training to transcutaneous electrical nerve stimulation. We judged that other terms used, including 'geriatric evaluation' and 'multidisciplinary geriatric assessment' returned articles much more cogent to the research question and made it unlikely that we would experience significant omissions by not including this term. From an initial reading of the articles retrieved (figure 1), four were identified as providing a particularly detailed articulation of the use of CGA in long-term care[24–27] and therefore provided a justifiable basis for initial programme theory development. These four articles were supplemented by semi-structured interviews that focused on how CGA was thought to work and what needed to be in place for it to achieve the outcomes of interest. Experts were purposively recruited on the basis of their knowledge of the care home setting and particular professional or occupational expertise to identify and test the range of assumptions about why and how CGA is believed to work and why it might be needed for UK care homes.

The data from the papers and interviews were analysed to identify possible configurations of the circumstances (*context*), in which clinicians, residents and care home staff respond to using CGA by changing their reasoning and behaviour (*mechanism*), which then led to changes in care for the residents (*outcome*). These were initially framed as 'if… then…' statements, which comprised initial programme theories of cause and effect.

Stage 2 involved testing these initial programme theories by revisiting the articles selected from the initial literature and grey literature searches. We looked for evidence that was relevant to how CGA was theorised to work in care home settings. Discussions among the team considered relevant evidence (eg, how clinical work is delegated and organised in the care home environment). Three

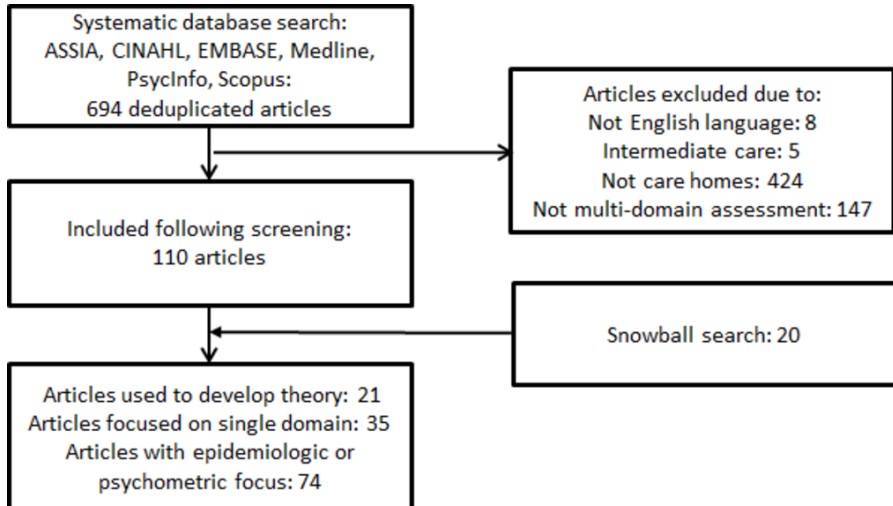

**Figure 1** Schematic of flow of articles within review.

reviewers (NC, MZ, LS) reviewed and discussed papers using qualitative data analysis software (NVivo V.11) to thematically code according to context-mechanism-outcome (CMO) configurations that were thought to explain how CGA worked in long-term care settings. Mechanisms were subdivided, as described by Dalkin et al,[28] into 'resource' and 'reasoning' components. Resources and reasoning are mutually constitutive of a mechanism but disaggregating them enables a better understanding of how resources work within a context, to change individual or group reasoning, to realise an outcome. As part of this stage, snowball searches—pursuing the references of references—were conducted, where it was felt likely from the text of the articles that particular articles would further inform the CMO configurations.[29]

Stage 3 involved discussion to develop the final iteration of the programme theory, a series of presentations to the whole project team and to the expert project steering group. The steering group comprised two lay representatives, two geriatricians, a general practitioner (GP), a representative from a national care home organisation and a social scientist with expertise in realist methodology. Three meetings were held with the project steering group and six with the project team. Presentations were structured to facilitate discussion of the plausibility of the programme theory, the supporting evidence and possible variations and/or alternative explanations.

### Patient and public involvement
One lay representative (family carer) was involved in designing the study and discussed resonance of findings with their lived experience. Early findings were presented to two dementia research patient and public involvement groups, two local GP patient participation groups and at engagement meetings held in three care homes which included residents and family members.

### Ethical approval
This realist review was a component of the PEACH project. The study protocols have been reviewed as part of good governance by the Nottinghamshire Healthcare Foundation Trust. Stakeholder practitioners who were invited to be interviewed were given participant information sheets and asked for their informed consent prior to interview.

## RESULTS
Following removal of duplicate articles, 694 articles describing CGA in older people's care homes were screened by title and abstract by three reviewers (NC, MZ, LS), and 110 were included in the review (figure 1). A further 20 articles were retrieved by iterative snowball literature search during stage 2. Nine interviews were held with stakeholders comprising a care home manager, an occupational therapist, a GP, four community geriatricians, an old age psychiatrist and a social worker.

Following the initial analysis of the four articles which provided detailed description of CGA in care homes, and the transcripts of scoping interviews during stage 1, a series of programme theories, comprising 'if… then…' statements, was written (box 1).

---

**Box 1  'If' and 'then' statements following stakeholder interviews and preliminary scoping of the literature**

*If* processes and structures (eg, information technology systems, agreed assessment protocols and opportunities to meet) are in place to facilitate a multidomain assessment of all needs of the individual resident, *then* this will lead to new insights and identification of care priorities and gaps in knowledge about an individual's status and needs.

*If* there are opportunities for all parties to discuss the significance of the comprehensive assessment data and how the resident's needs are addressed, *then* they will have a common understanding of the resident's situation and address and prioritise person-centred goals.

*If* practitioners are able to use the assessment data to agree shared goals for the individual, and the care plan describes care activities and treatments which may lead towards such goals, *then* the resident's treatment is personalised leading to a good experience of care, staff satisfaction and reduced need for acute or crisis services.

---

While the 'if… then…' statements were broad, they aligned how CGA works with literature on interprofessional working,[30] a recognition in the care home research of the intrinsic power imbalances between health and social care staff during encounters[31] and the importance of delivering minimally disruptive medicine to a patient group who do not fit acute and primary care models of care.[32] They also reflected three of the components previously described as constituting CGA, namely structured comprehensive assessment, developing a plan of care and working towards the delivery of patient-centred goals.[13]

Iterative review of the wider literature enabled us to further refine our initial programme theories to delineate context, mechanism and outcome configurations. We were able to address the first two objectives—identifying the elements which made CGA effective and considering how context influences the implementation, uptake and working of CGA in care homes—but there were no studies that explicitly addressed the feasibility of CGA in UK care home setting and we were therefore unable to address our third study objective regarding this. There were no studies which focused primarily on HRQoL but evidence that delivery of CGA was associated with improved outcomes related to resident satisfaction,[33] healthcare resource use,[34] reduction of polypharmacy[35] and objective measures of quality of care.[27] See online supplementary appendix 2 for summary of all articles which informed the development and testing of theory.

The programme theories for the three components of CGA are described below and summarised in boxes 2-4. Due to the information flow between these three components, they build on each other in a way which is summarised in figure 2.

### Structured comprehensive assessment (CMO 1)

The practices and processes of assessment inform the whole CGA (box 2). Assessment of many domains of health status and impairments was represented as crucial in building a picture of an individual's complex needs and views about their personal priorities and goals.[36] Unlike

---

**Box 2  Context-mechanism-outcome configuration for structured comprehensive assessment**

*Context*: a structured or standardised process, supported by a multidisciplinary care team, with resources available to conduct assessments with the resident.
*Mechanism resources*: protocols, tools and communication systems are ready-at-hand to enable information about an individual's health status and personal wishes to be recorded and shared with the multidisciplinary team. Information given by the spouse or family member of the resident, or by care workers may also be drawn upon, particularly when an individual has limited capacity.
*Mechanism reasoning*: reframing multiple accounts into a comprehensive representation of an individual's health status rather than a description of syndromes.
Outcomes: overview of needs and goals of resident. Availability of information for transfer to another setting.

---

**Box 3  Context-mechanism-outcome configuration for developing a care plan**

*Context*: communication between multidisciplinary practitioner team members from different organisations.
*Mechanism resources*: case-conference discussion of the comprehensive structured assessment to gain an understanding of aims and priorities of individual.
*Mechanism reasoning*: developing a unified view and shared aims and goals for the resident.
*Outcome*: a comprehensive care plan is developed and prescriptions updated.

---

discipline-specific needs assessment that may focus on a particular syndrome or care pathway, structured comprehensive assessment requires an overview of all domains.[37]

The context of this component is the agreement and support of health and social care practitioners, including care home staff, to work in a coordinated and collaborative team. Furthermore, the context includes resources to enable the team to collaborate, including staff time and agreements on information exchange. Unlike hospital-based studies[7–9] that can assume the presence of a geriatrician and a predictable (therapist, nurse, social worker) multidisciplinary team (MDT), there was evidence of greater variability in the professional groupings comprising the core MDT for CGA in the care home setting. Team members are likely to represent different provider organisations, and so strategic collaboration between organisations is another important context. Within the studies included in this review, there was variability in the extent to which individual professionals making up the MDT were available. In some studies, specific personnel were hired to deliver CGA.[12] Descriptions of team composition[27 34 38 39] most frequently described a team comprising social carers, nurses and a physician, with physiotherapists, occupational therapists or pharmacists added to this in some situations. Bellantonio *et al* described a large team consisting of a geriatrician, advanced nurse practitioner, physical therapist, dietitian and social worker.[34] The included articles do not specify that geriatric training or qualification is

---

**Box 4  Context-mechanism-outcome configuration for working towards the delivery of patient-centred goals**

*Context*: coordination of care activities.
*Mechanism resources*: resident may be referred to topic-specialist practitioners to give further assessment and together with team members refine the care plan.
*Mechanism reasoning*: delegation from the multidisciplinary team to care home staff provides authority to deliver care according to the care plan.
*Outcome*: resident perceives continuity of care and early intervention either prevents deterioration of needs or provides a palliative response. Unnecessary care is avoided.

---

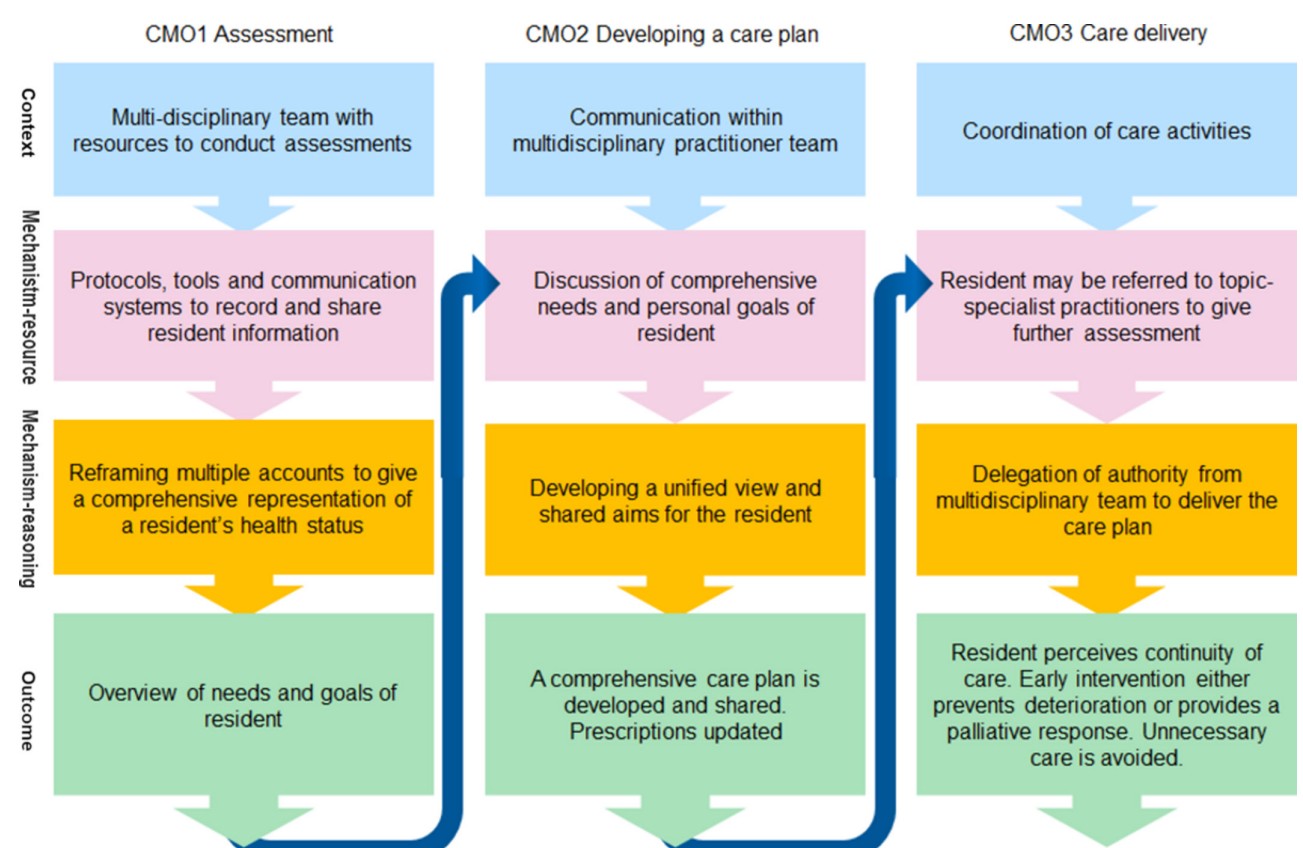

**Figure 2** Nested arrangement of context-mechanism-outcome (CMO) configurations.

a requirement for CGA, however geriatric expertise is mentioned in two articles.[26 38]

The mechanism resources of the structured assessment are the protocols and various tools that are administered while assessing each domain of CGA, such as the mininutrition assessment and Barthel Index of activities of daily living (function).[33 38] The Minimum Data Set (associated with the InterRAI system) has created a standardised battery of tools and developed a computer-aided assessment protocol.[40] We included articles describing MDS/InterRAI because it is described as a comprehensive assessment which supports an interdisciplinary team.[41] Here we interpret that, in functioning to manage the care of an individual, InterRAI could be described as a technological development of CGA that structures rather than triggers the assessment process. The computer-facilitated multidomain assessment is a mechanism resource which enables an individual practitioner, often a nurse, to conduct the initial assessments of the resident.[36] The computer-based algorithm may prompt further assessment by other members of the MDT.[27] The literature to date does not include sufficient evidence to suggest distinct mechanisms, and therefore we argue that the computer-facilitated assessment and the conventional CGA assessment can be described by the same programme theory (box 2).

Facilitated by either conventional notes and assessment tools, or through computer-aided collection of Minimum Data Set information, the mechanism reasoning of the

MDT is to reframe multiple accounts into a systematic representation of an individual's health status.[12] This is in contrast to a conventional needs assessment which may focus on one domain or syndrome, assessed by one professional.

> Taking stock of all observations, information and knowledge about a resident, and understanding the resident's limitations and strengths.[36]

The emphasis in the literature was on policy and practice development as an agent of change for care, with the result that staff became more person-centred and better understood the different care needs of the care home population. The processes of care driven by these contexts were not, however, well described. Studies described staff listening to goals and expectations of residents in order to attend to the identity of the whole person rather than a list of diagnostic categories.[24] One paper also described collecting family members' views, using these largely as a source of information during multidisciplinary meetings.[34] With the standardised assessment, one study described how the third iteration of the US minimum dataset (MDS3) instrument had been influenced by person-centred policy context by including, within the questionnaire, opportunities to ask the resident open questions and explore 'subjective states' such as pain, mood and cognitive functioning.[42] Bringing a focus on the views of the individual resident may have changed attitudes within the MDT to create a

mechanism of common understanding of the importance of knowing the resident, however there was little evidence within the selected literature to support this mechanism.

The outcome of this component of CGA is primarily an improvement in quality of information about the individual resident which is shared and reviewed over time among the MDT.

### Developing a care plan (CMO 2)

The second component of CGA (box 3) is about communication between practitioners that creates a shared view of the priorities for delivery of care. The literature suggests that this part of the process is more focused around interprofessional communication, as involvement of residents or their families is not explicitly described.[43] The information generated by structured comprehensive assessment about the needs and goals is the input for the multidisciplinary discussion and thus the outcome of CMO 1 becomes a resource mechanism for CMO 2.

The context for the second component is a structure for communication between the team members in order to interpret a resident's needs assessments.[38] Practitioners may be based at a distant location from the care home and therefore, to enable practitioners to work together, time needs to be set aside for case conference meetings. The resources required for coordination and administration of an MDT (and associated costs) are not described in the articles.

A Dutch trial reported that prior to the CGA intervention, attendance of GPs at the MDT meeting was infrequent (25%) and, during the trial, this increased to 90%.[27] Availability of practitioners to discuss the assessment findings is the mechanism resource which facilitates communication between members of the team, for example, in a case-conference meeting.[39] This meeting may occur regularly, for example, monthly.[27] Most studies did not explicitly state whether the resident or family were invited to or attended team meetings and, where they did attend there was no mention of them doing much more than providing information. The members of the meeting included nurses, primary care doctors and care home managers. In some studies, geriatricians or therapists were also specified as part of the team.[27 34]

Different professional or disciplinary perspectives on the needs and care requirements of the individual may be discussed in MDT communications. The team discussion prompts the mechanism reasoning which is to create a unified view of priorities for the care plan for the individual.[26] Topic specialists may contribute to assessment and pool their expertise with the team, facilitating working towards common goals for the individual resident, or continuity of care.[38]

> …trained professionals who design a shared disease management plan[27]

The outcome of practitioner teamwork is to develop a comprehensive care plan. An immediate outcome may be referrals to other practitioners, external to the team. One study, for example, showed an increase in referrals to allied health practitioners, hospital at home service and palliative care.[33] The same study found small decreases in hospital readmission rates, suggesting that these community referrals may mitigate against hospital attendance. Changes in prescribing were also reported as an important immediate output from the second component of CGA.[39]

### Working towards the delivery of patient-centred goals (CMO 3)

The outcome of the comprehensive care plan becomes the resource mechanism for the next stage of CGA, which focusses on working towards the delivery of patient-centred goals.[44] It was often explicit within articles that a care plan was implemented as part of CGA but there was less emphasis on standardisation at this stage than during structured comprehensive assessment. There was at times a recognition of the need to file the outputs of the assessment process but it was not made explicit whether these were in the form of structured assessment outputs or specific care plans. An example is the 'Care by Design' programme, reported by three Canadian articles within our review, which specified inclusion of a summary document of assessment information in the resident's medical chart.[12 26 35] The authors describe the expectation that this summary sheet would be considered by all care staff and transferred with the resident to urgent care hospital. However, the form appears to be the same form which has been used to collect the summary comprehensive assessment information, rather than a care plan generated out of multidisciplinary discussion.

Three articles described care planning within the InterRAI algorithm-driven system. This has a series of predesigned resident assessment protocols that lead to comprehensive care plans. The latter appear to address syndromes or diseases, rather than being truly comprehensive.[45] Two articles described problems perceived by care assistants when implementing care plans, suggesting either a lack of resources or an emphasis on a systematic approach over individualisation; care assistants reported feeling disempowered and uninvolved in decision making.[46 47]

While the specific format of the care plan is not described in detail, the outcome is delivery of care activities in a coordinated way. Empirical outcomes include improvements in quality of care, and reduced transfers to hospital for urgent care.[27 33 34]

### A programme theory of CGA in care homes

Figure 2 outlines the nested configurations of CMOs for CGA in care homes, where outcomes of earlier stages form the mechanism resource for later stages. This makes it clear how each CMO builds on the one before.

The delivery of CGA in care homes was dependent on two contexts, how feasible it was for professionals to participate and whether the use of a structured assessment created the sense of a common purpose challenging

discipline-specific approaches and behaviours. We heard initial suggestions from our stakeholders that the CGA process could mitigate differences in disciplinary status to enable more balanced power relationship between all clinicians, social workers and care home staff. However, we found minimal explicit statements within the selected evidence to suggest that this had been considered.

Several articles support the outcomes of CGA in terms of health benefit and delivery of healthcare services. Responses from resident and family perspectives indicated a positive experience or satisfaction with services.[25 33] Practitioner team members also reported positive outcomes of CGA, although for programmes that involved computerised information, care home staff identified a challenge with access to computers to enter data and read care plans, meaning that staff felt excluded from sharing views on care plans.[46]

## DISCUSSION

We were able to find evidence of how comprehensive geriatric assessment operates in care homes and we have developed a programme theory demonstrating how the processes identified map to the way in which CGA is described to take place in hospital and community settings. Effective delivery of CGA requires a structured or standardised approach to assessment, followed by communication within an MDT and coordination of care delivery activities. The evidence suggests that for CGA to be effective all three of these components must be present.

The strength of this realist review is in the structured approach, with putative 'if… then…' statements shaped by insights from stakeholder interviews, followed by systematic literature search and iterative development of programme theory guided by frequent and repeated recourse to the published literature. The resulting programme theory is based on insights from multiple studies where consistent patterns of association between context, mechanisms and outcomes have been found. By using scoping interviews and an expert project-steering group, we were able to make the greatest possible sense of the limited literature in a structured and robust way.

A significant limitation of the work is that, by limiting the studies to those which explicitly used CGA as a descriptor, we may have missed out research using other conceptual frameworks to describe multidisciplinary and integrated care in care homes. It is possible that these frameworks may have overlapped sufficiently with CGA and would have shone further light on our theoretical framework had we extended our search to include them. We defend the decision not to include them because interventions which may have similar aims, but which do not comply with our stated characteristics of CGA are unlikely to have identical mechanisms, if considered from a realist evaluation perspective. The weight of evidence was greatest for processes of assessment, while less data were available on processes of developing a care plan and working towards patient-centred care.

To our knowledge, this is the first realist review of CGA in care homes. A recent systematic review summarised characteristics of different assessment tools used to support the CGA process, but did not synthesise the processes and relationships.[38] This current review builds on that research by providing greater detail of how CGA works in care homes, and under what circumstances.

The programme theory provides a framework which will be of use to providers developing, implementing and evaluating the efficacy of CGA. It also has some explanatory value when considering why interventions aiming to coordinate care in care homes using assessment protocols or pathways have not always been shown to be effective.[48 49] Our realist review suggests that the engagement of an MDT with regular communication to enable development of a care plan, followed by coordinated delivery of this, is at least as important as the assessment frameworks used.

An important finding was the tension between standardised forms of assessment (MDS or InterRAI), compared with the development of tailored or person-centred care plans in both InterRAI and conventional CGA.[50] The theory of competing institutional logics, of standardisation and customisation,[51] may help to interpret this tension. For the standardised assessment (MDS), the aim is to generate standardised data in order to be comparable for analysis, however customisation is required to enable the MDT to develop a care plan to meet the complex needs of the individual. Good communication within the MDT is important to address this tension between standardisation and customisation. Indeed, a previous realist review of integrated care programmes in the community identified trust as the important mechanism leading to outcomes of improved patient experience and health.[52] The idea that CGA provides an opportunity where professionals from different teams and organisations can come together with residents to establish shared priorities for healthcare of a resident is similar to the concept of 'communicative space' described elsewhere in the literature.[53] This has been suggested to be particularly important in care homes because they are contested places where neither professionals nor residents may feel fully in charge of events.

Despite limiting our search to literature on care homes, there was a lack of explicit description about how CGA was tailored specifically to the care home setting. For example, there were limited descriptions of how staff and other resources should be organised to optimise CGA within care homes. There was little description of the training or competencies required to enable CGA in a sector of health and social care where it is not routinely delivered. This made it difficult to address our third study objective around the feasibility of CGA in UK care homes. Further work is required to better understand the care home-specific aspects of CGA, which are essential to guide new practitioners as they attempt to implement CGA in the sector.

## CONCLUSION

In summary, this work provides a programme theory which describes how CGA works in care homes. CGA has three components: a structured or standardised approach to assessment, followed by communication within an MDT and coordination of care delivery activities. Unless all three of these components are addressed, CGA is unlikely to be successful in this setting. This new understanding provides the basis for more evidence-based approaches to service development and audit in the field. The programme theory provides the basis for further research around the implementation of CGA in this sector.

**Author affiliations**
¹Division of Rehabilitation and Ageing, School of Medicine, University of Nottingham, Nottingham, UK
²National Institute of Health Research Collaboration for Leadership in Applied Health Research and Care East Midlands, Nottingham, UK
³Centre for Research in Public Health and Community Care, University of Hertfordshire, Hatfield, UK
⁴National Institute of Health Research Collaboration for Leadership in Applied Health Research and Care East of England, Cambridge, UK
⁵Faculty of Health, Psychology and Social Care, Manchester Metropolitan University, Manchester, UK
⁶Santa Maria University Hospital, Lisbon, Portugal
⁷Nottingham Biomedical Research Centre, University of Nottingham, Nottingham, UK
⁸Division of Psychiatry and Applied Psychology, School of Medicine, University of Nottingham, Nottingham, UK
⁹Division of Medical Sciences and Graduate Entry Medicine, School of Medicine, University of Nottingham, Nottingham, UK
¹⁰School of Health Sciences, City, University of London, London, UK

**Acknowledgements** The PEACH study research team are, in addition to the authors: Mr Zimran Alam, Ms Anita Astle, Professor Tony Avery, Dr Jaydip Banerjee, Professor Clive Bowman, Mr Michael Crossley, Dr Reena Devi, Professor Heather Gage, Dr Kathryn Hinsliff-Smith, Ms Gemma Housely, Dr Jake Jordan, Dr Sarah Lewis, Professor Pip Logan, Ms Annabelle Long, Professor Finbarr Martin, Professor Julienne Meyer, Dr Dominick Shaw, Professor David Stott and Ms Adeela Usman.

**Contributors** NC, MZ and LS carried out searches, data extraction and theory was developed together with CG and AG. NC, CG and AG conducted synthesis with reflections and amendments from JRFG, TD and other authors. All authors have read and approved the final version.

**Funding** This work has been undertaken as part of the Proactive Healthcare in Care Homes (PEACH) study, funded by the Dunhill Medical Trust, award no FOP1/0115.

**Competing interests** None declared.

**Patient consent for publication** Not required.

**Ethics approval** The protocol for the whole project, including the realist review, was submitted to UK Health Research Authority and University of Nottingham Research Ethics Committee; these committees identified the project as service development.

**Provenance and peer review** Not commissioned; externally peer reviewed.

**Data sharing statement** This is a review of published or open access literature.

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
