## [Reviewer comments · BMJ Open]

ARTICLE DETAILS

TITLE (PROVISIONAL)	The Role of Comprehensive Geriatric Assessment in Healthcare of Older People in UK Care Homes: realist review
AUTHORS	Chadborn, Neil; Goodman, Claire; Zubair, Maria; Sousa, Lidia; Gladman, John; Denning, Tom; Gordon, Adam

VERSION 1 - REVIEW

REVIEWER	Graham Ellis Monklands Hospital NHS Lanarkshire Scotland
REVIEW RETURNED	06-Nov-2018

GENERAL COMMENTS	This paper is excellent and I really have little to add or quibble about. Firstly it is a new area for CGA research and there is little published to date that addresses how CGA should be addressed in care homes. I suspect it may well form a background to further work and be well cited. The format of the research is appropriate to context as theory generating and descriptive. The unpacking of the elements of CGA is excellent - concise and logical and fits with the literature. A really helpful step forward. my only questions would be: CGA technically specialist delivered (i.e. by a geriatrician) so although the approach is correctly identified as a CGA approach, I wonder if more discussion is worth considering on the balance between a specialist or generalist approach. It does not alter the conclusions and the authors correctly identify I think that context is what matters in a care home setting. Secondly one or two of the references are a little out of date. Number 8 is superseded by a new Cochrane review 2017 on the Cochrane library which also references some of the material covered in reference 12. I have no other comments and think this paper deserves publication.
--

REVIEWER	Julie Whitney King's College London
REVIEW RETURNED	05-Dec-2018

GENERAL COMMENTS	This is a well written paper. It is a valuable subject which needs further exploration and the technique used was an appropriate way to address the research questions. The paper demonstrates a very good addition to understanding of the issues relating to CGA in care homes.
---

	I am not an expert on the realist reviews and therefore cannot comment with authority on the method, other than to say that it appears to be sound. One minor comment - would be to suggest that the first time the abbreviation CMO is used, it is defined. I know that this is already described in the method, but this would be helpful for readers with little experience of this research method.
--	---

REVIEWER	Stephanie Harrison South Australian Health and Medical Research Institute, Australia
REVIEW RETURNED	09-Jan-2019

GENERAL COMMENTS	This is a well written paper discussing an important topic of the role of CGA in care homes using a realist review approach. I have some considerations and questions for the authors:  -I would like the authors to include the RAMESES list of items to be included when reporting a realist synthesis with a description for each point to show where in the paper the authors have addressed the items -the authors state the mesh term 'Geriatric Assessment' was not included because it resulted in too many articles-how many was this and how confident are the authors that relevant texts were not omitted because of this -Please include the date of the search of electronic databases -The authors state that: 'A significant limitation of the work is that, by limiting the studies to those which explicitly used CGA as a descriptor, we may have missed out research using other conceptual frameworks to describe multi-disciplinary and integrated care in care homes. It is possible that these frameworks may have overlapped sufficiently with CGA and would have shone further light on our theoretical framework had we extended our search to include them.' why was this set as a limitation and I imagine this will limit the amount of international evidence obtained, how do the authors justify this? -the authors state there were 'limited descriptions of how staff and other resources should be organised to optimise CGA within care' and 'Information given by the spouse or family member of the resident, or by care-workers may also be drawn upon, particularly when an individual has limited capacity.' Is this not critical for individuals with high levels of cognitive impairment? did the literature not specifically discuss involving family members for people with high cognitive impairment or other limitations which would limit their involvement in a CGA? I think this needs to be highlighted as an area for future research if the literature is not adequately describing the benefits of families and resident participation and resident/family participation should be considered in Figure 2 -In the appendix please provide titles for the tables
--

VERSION 1 – AUTHOR RESPONSE

Reviewer 1

The question of specialist input or leadership of the CGA process is very pertinent, and yet there is little specific guidance on this issue within the literature. Several articles mention geriatrician input,

into the MDT, so this may be expected to be an element of CGA. Two articles mention expertise in caring for older people and training. We have added the following text:

“The included articles do not specify that geriatric training or qualification is a requirement for CGA, however geriatric expertise is mentioned in two articles (26, 39).”

We thank the reviewer for highlighting the more recent Cochrane review of hospital based CGA. We have largely used the previous review for its descriptive content of the process of CGA, rather than the specific outcomes, because it is describing the hospital rather than care home setting. However it is good practice to also reference the update of the review, so we will include this.

Reviewer 2

We thank the reviewer for pointing out the lack of clarity about describing CMO initially within results section. We have amended this as follows, eg the first subheading of the results section:

“Structured comprehensive assessment (Context Mechanism Outcome (CMO) 1)

Reviewer 3

We acknowledge that choice of search terms is a judgement of sensitivity and selectivity; we felt that the MeSH term “Geriatric Assessment” was not selective enough and would lead to substantial time invested in screening rather than interpreting articles that had already been retrieved through more selective searches. We have amended the text to clarify:

“The only deviation from the published protocol was that we did not use the Medical Subject Heading (MeSH) term “Geriatric Assessment” in Pubmed. This was because we found this term to be highly inclusive, returning articles on topics ranging from posture and mobility training to transcutaneous electrical nerve stimulation. We judged that other terms used, including “geriatric evaluation” and “multidisciplinary geriatric assessment” returned articles much more cogent to the research question and made it unlikely that we would experience significant omissions by not including this term.”

We have now included the date of the database searches:

“Databases were searched initially on 19th May 2017 and finally on 12th July 2018.”

We acknowledge the challenge of defining the scope and definition of complex interventions such as CGA, and have added a justification as follows:

“We defend the decision not to include them [similar interventions that did not use the term CGA] because interventions which may have similar aims, but which do not comply with our stated characteristics of CGA are unlikely to have identical mechanisms, if considered from a realist evaluation perspective.”

Some of the included articles did mention the resident or family perspective, and we mention the individual and families input in CMO1. There was a lack of specific description of how this would occur apart from questions in MDS3 exploring “subjective states” such as pain, mood and cognitive functioning (Reference 43, Thomas et al 2014). Cognitive impairment was not discussed explicitly and we assume that addressing cognitive impairment and communication difficulties was implicit within the professional practice of the MDT. We agree that these points are worthy of future research.

VERSION 2 – REVIEW

REVIEWER	Stephanie Harrison South Australian Health and Medical Research Institute, South Australia
REVIEW RETURNED	24-Jan-2019

GENERAL COMMENTS	Thank you for addressing my comments.
---------------------------------------